# A cell cycle-dependent CRISPR-Cas9 activation system based on an anti-CRISPR protein shows improved genome editing accuracy

Daisuke Matsumoto[1,3], Hirokazu Tamamura[1] & Wataru Nomura [1,2✉]

The development of genome editing systems based on the Cas9 endonuclease has greatly facilitated gene knockouts and targeted genetic alterations. Precise editing of target genes without off-target effects is crucial to prevent adverse effects in clinical applications. Although several methods have been reported to result in less off-target effects associated with the CRISPR technology, these often exhibit lower editing efficiency. Therefore, efficient, accurate, and innocuous CRISPR technology is still required. Anti-CRISPR proteins are natural inhibitors of CRISPR-Cas systems derived from bacteriophages. Here, the anti-CRISPR protein, AcrIIA4, was fused with the N terminal region of human Cdt1 that is degraded specifically in S and $G_2$, the phases of the cell cycle when homology-directed repair (HDR) is dominant. Co-expression of SpyCas9 and AcrIIA4-Cdt1 not only increases the frequency of HDR but also suppress off-targets effects. Thus, the combination of SpyCas9 and AcrIIA4-Cdt1 is a cell cycle-dependent Cas9 activation system for accurate and efficient genome editing.

[1] Institute of Biomaterials and Bioengineering, Tokyo Medical and Dental University, 2-3-10 Kandasurugadai, Chiyoda-ku, Tokyo 101-0062, Japan. [2] Graduate School of Biomedical and Health Sciences, Hiroshima University, 1-2-3 Kasumi Minami-ku, Hiroshima 734-8553, Japan. [3] Present address: Daisuke Matsumoto, Department of Chemistry, The Scripps Research Institute, La Jolla, CA 92037, USA. ✉email: wnomura@hiroshima-u.ac.jp

The CRISPR-Cas9 system was originally discovered as part of the bacterial immune system against external DNA from organisms such as bacteriophages and plasmids[1–3]. It has become the predilected simplified genome editing tool, because it is easier and less expensive to construct various target libraries compared to other editing technologies such as ZFN and TALEN. The system has shown its efficient editing when used in human cells[4–6] and model organisms[4,7–13]. The wide application of CRISPR technology is expected in the fields of agriculture, medicine, biotechnology and others in the coming years. Although CRISPR technology is the most useful method for genome editing, off-target effects that cause unexpected mutations at pseudo-target DNA sequences could occur, similarly to those seen using as ZFN and TALEN. Thus, in addition to endeavor to increase the efficiency of precise editing at on-targets, off-target effects should be carefully addressed when genome editing tools are used, especially for clinical applications.

Homology-directed Repair (HDR) is a precise DNA repair pathway based on template DNA having homologous arm sequences adjacent to the cleavage site. In HDR events, repair of target sequences introduces precise mutations. However, depending on the length of the homologous arm, off-target sites may not be recognized for editing through HDR. Therefore, increasing the ratio of DNA repair through HDR over non-homologous end joining (NHEJ) is important for precise genome editing. These two repair processes show different cell-cycle dependency. HDR occurs during the S and $G_2$ phases, whereas NHEJ operates in all phases of the cell cycle, especially in $G_1$[14]. It has been reported that the efficiency of genome editing through HDR is influenced by chemical or genetic disruption of the NHEJ pathway[15,16]. The efficiency of HDR can also be increased by controlling the timing of the delivery of SpyCas9-single guide RNA (sgRNA) ribonucleoprotein (RNP) complexes to chemically synchronized cells[17]. However, cytotoxicity to cells could be a concern when chemicals are used to interrupt the NHEJ pathway or synchronize cell-cycle. Activation of Cas9 endonuclease specifically during S and $G_2$ phases by fusing with the N-terminus of Geminin (1–110) has been reported[18,19]. The method could provide a way for avoiding cytotoxicity. However, HDR activity was only marginally increased, which was probably due to the amount of Cas9-Geminin (1–110) fusion was not fully recovered in the S phase after degradation in the $G_1$ phase.

Recently, anti-CRISPR (Acr) protein inhibitors of the CRISPR-Cas9 system have been found[20–24]. The inhibitors, including AcrIIA4, were derived from bacteriophages targeting pathogenic bacterial strains. AcrIIA4 from *Listeria monocytogenes* prophage binds strongly to SpyCas9-sgRNA complexes ($K_D = 0.6$ nM), but the binding affinity to ApoSpyCas9 is lower ($K_D = \sim 4.8$ μM)[25]. It has been reported that AcrIIA4 efficiently inhibits SpyCas9 activity in mammalian cells[26,27]. Furthermore, the inhibition of SpyCas9 activity by AcrIIA4 reduces off-target editing[21]. Thus, when anti-CRISPR expression can be controlled by cell cycle, the activity of Cas9 endonuclease could also be controlled in the cells. Here, we fused the anti-CRISPR AcrIIA4 with the N-terminal region of human chromatin licensing and DNA replication factor 1 (hCdt1) for activation in the S/$G_2$ phases and inactivation in the $G_1$ phase. hCdt1 is degraded by ubiquitin-mediated proteolysis through the SCF$^{Skp2}$ complex in the S/$G_2$ phases. The cell cycle dependent Cas9 activation system was validated using SpyCas9 endonuclease and AcrIIA4 in the cells. As expected, the system displayed autonomous Cas9 activity switch dependent on the cell cycle.

## Results

**Construction of cell cycle dependent expressing anti-CRISPR protein**. Genomic DNA replication occurs during the S phase and is strictly controlled by licensing machinery in the cells[26,27]. Cdt1 is a protein that acts as a licensing factor, preventing over-replication in higher eukaryotes. The function of Cdt1 is inhibited by degradation through ubiquitin mediated proteolysis[28,29] and Geminin binding[30,31]. The N-terminus of Cdt1 is a ubiquitylation domain targeted by two E3 ubiquitin ligases, CUL4$^{Ddb1}$ (Cullin 4, damage-specific DNA-binding protein 1) E3 ligase and SCF$^{Skp2}$ E3 ligase[32]. SCF$^{Skp2}$ E3 ligase targets phosphorylated amino acid(s) (Ser31 and/or Thr29) during the S and $G_2$ phases. Cyclin A-dependent kinases catalyze these phosphorylation reactions. The cyclin-binding motif (Cy motif), Arg68-Arg69-Leu70, of Cdt1 is required for the phosphorylation. Monomeric Kusabira-Orange (mKO) fluorescence protein fused with amino acids residues 30 to 120 of Cdt1(Cdt1(30–120)) has been developed[33]. This fusion protein, which is designated Fucci, can be used to visualize cell cycle phases. In the Fucci system, mKO2 is expressed in the $G_1$ phase and degraded in the S, $G_2$, and M phases by proteolysis mediated by SCF$^{Skp2}$ E3 ubiquitin ligase. Cell-cycle dependent expression of the Cdt1 domain fused with mKO2 fluorescent protein was observed by confocal laser scanning microscopy (CLSM), using the expression plasmid of Fucci. Time-lapse observation of mKO2 expression suggested that the expression

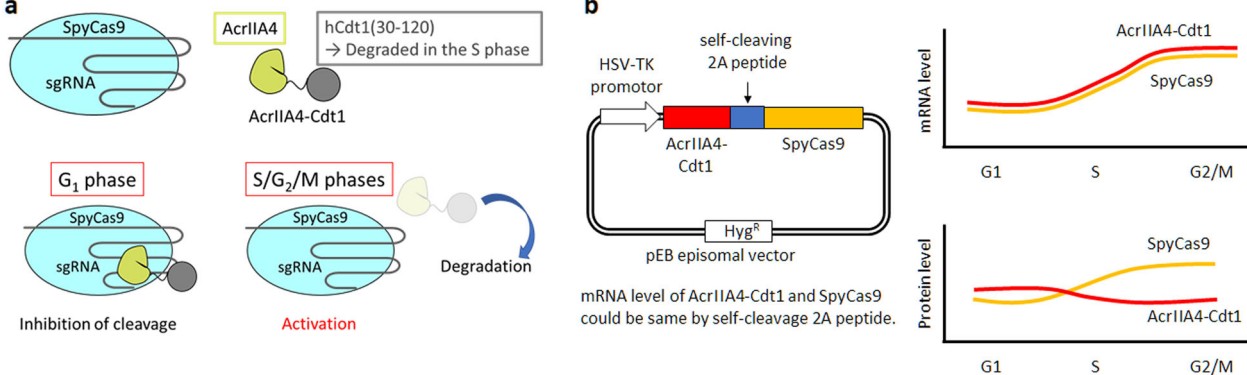

**Fig. 1 Anti-CRISPR mediated cell cycle specific Cas9 activation system. a** Description of anti-CRISPR mediated cell cycle specific Cas9 activation system. In $G_1$ phase, AcrIIA4 which is a known inhibitor of SpyCas9 inhibits Cas9-sgRNA by binding the complex. In S/$G_2$/M phases, AcrIIA4 is degraded because of S phase degradation domain from Cdt1, and the SpyCas9-sgRNA complex is activated. **b** Constructed episomal vector and hypothesized expression change of AcrIIA4-Cdt1 and SpyCas9.

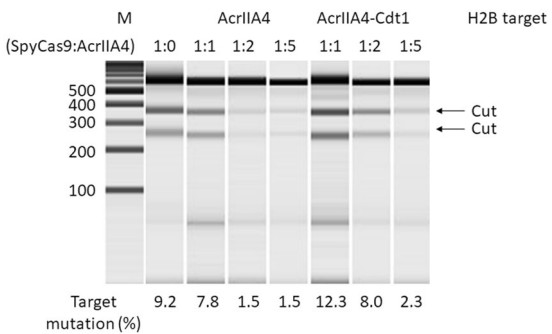

**Fig. 2 Inhibition of cleavage by SpyCas9 and guide RNA (gRNA) targeting *H2B* gene by using AcrIIA4-2A-Cas9 or AcrIIA4-Cdt1-2A-Cas9.** The molar ratio of plasmid (SpyCas9:AcrIIA4) changed from 1:1 to 1:5. The cleavage activity of SpyCas9 was calculated using the T7E1 assay.

level changed in a time-dependent manner (Supplementary Fig. 1). When AcrIIA4 is fused with Cdt1(30–120), designated as AcrIIA4-Cdt1, the fusion protein was expected to be degraded at the $S/G_2$ phases of the cell cycle because of ubiquitin-mediated proteolysis as shown by mKO2-Cdt1 fusion[34,35]. We hypothesized that when SpyCas9 is co-expressed with AcrIIA4-Cdt1 in the cells, SpyCas9 activity is inhibited during the $G_1$ phase, and is regained when AcrIIA4-Cdt1 is degraded in the $S/G_2$ phases. Therefore, HDR-mediated gene correction could be dominant (Fig. 1a, b). Plasmid DNA encoding AcrIIA4 or AcrIIA4-Cdt1(30–120) with the nuclear localization signal (NLS) was constructed and used for transient transfection to 293A cells. Localization of AcrIIA4 and AcrIIA4-Cdt1 in cell nuclei was confirmed by immunostaining 48 h after transfection (Supplementary Fig. 2). To examine whether the Cdt1(30–120) domain affected the inhibition activity of AcrIIA4, co-expression of SpyCas9, sgRNA targeting *H2B* gene, and AcrIIA4/AcrIIA4-Cdt1 was performed. Inhibition activity was evaluated by mutation analysis using the T7E1 assay. A decreased mutagenesis rate was observed when the molar ratio of AcrIIA4/AcrIIA4-Cdt1 to SpyCas9 was increased (Fig. 2).

**Construction of autonomous controllable CRISPR and effects on genome editing with donor plasmid for precise editing.** It was shown that the mutagenesis rate by NHEJ was decreased by AcrIIA4-Cdt1, and was almost completely suppressed by AcrIIA4 alone. To control the amount of SpyCas9 and AcrIIA4 DNA more precisely, plasmid DNA encoding AcrIIA4-Cdt1 and SpyCas9 separated by a self-cleaving peptide sequence (T2A) was constructed (Fig. 1b). We hypothesized that the amounts of AcrIIA4-Cdt1 and SpyCas9 were strictly regulated, as in previous reports using T2A peptide[36–39]. Although a proline residue is added to the N-terminus of SpyCas9 after cleavage of T2A, SpyCas9 activity should not be affected by this, as the N-terminus of SpyCas9 is exposed to the surface in the apo form or in the complex with guide RNA and target DNA[40]. As the expression plasmid was changed to an episomal vector, the encoded proteins could be stably expressed without gene integration into the host genome. Cell cycle-dependent expression of the AcrIIA4-Cdt1 fusion protein was firstly analyzed. After transient transfection of the plasmid encoding AcrIIA4-Cdt1-2A-Cas9, cells were selected by Hygromycin. The surviving cells were treated with Thymidine or Nocodazole for synchronization. Cells obtained after release from drug treatment were analyzed by fluorescence-activated cell sorting (FACS) (Supplementary Fig. 3) and the expression levels of

AcrIIA4, AcrIIA4-Cdt1, and SpyCas9 were checked by western blotting (Fig. 3). Decreased expression of AcrIIA4-Cdt1 fusion was observed at the $S/G_2/M$ phases, while it was increased at the $G_1$ phase. The change of expression level was about 3-fold. The results confirm that the change of expression level of AcrIIA4-Cdt1 fusion depends on the cell cycle, as observed in the mKO2-Cdt1 (Fucci) expression analysis (Supplementary Fig. 1). The expression levels of AcrIIA4 without the Cdt1 domain did not change depending on the cell cycle. In addition, SpyCas9 expressed from the same gene cassette displayed no correlation in the expression levels with the cell cycle. Decrease of mutagenesis rates by NHEJ were observed when using AcrIIA4-2A-Cas9 (0.9%) and AcrIIA4-Cdt1-2A-Cas9 (3.7%) expressing vectors compared to that of SpyCas9 vectors alone (8.0%) (Fig. 4). The difference in decreased NHEJ rates between AcrIIA4 and AcrIIA4-Cdt1 suggests that the fusion is limiting the activity of AcrIIA4 by the cell-cycle dependent degradation of Cdt1. Expression of AcrIIA4-Cdt1 fusion showed higher mutagenesis rates compared to that of AcrIIA4 alone. This suggests that cell-cycle dependent expression of AcrIIA4-Cdt1 could inhibit SpyCas9 in the $G_1$ phase and release it in the other phases while AcrIIA4 continuously inhibits SpyCas9.

Precision editing using donor plasmid and AcrIIA4-Cdt1 was examined with the AcrIIA4-Cdt1-2A-Cas9 expression plasmid. After transient transfection of the episomal vector for SpyCas9 and AcrIIA4/AcrIIA4-Cdt1, cells were selected using Hygromycin for 3-7 d. The surviving cells were transfected with plasmid coding sgRNA targeting the *AAVS1* site and donor plasmid by lipofection. The donor plasmid encoded two recognition sites for sgRNA-SpyCas9, so the plasmid would be cleaved in the cells and double stranded repair fragments would be formed[41]. Insertion of the repair DNA sequence was confirmed by XhoI digestion 72 h after transfection (Fig. 5a). Specific XhoI digestion indicating insertion of repair sequence was not observed in the absence of the sgRNA, nor in the case when AcrIIA4 and SpyCas9 were co-expressed. In the presence of SpyCas9 alone, XhoI digestion bands were observed and the efficiency of precision editing was estimated to be 1.6%. In the case of co-expression of AcrIIA4-Cdt1 and SpyCas9, efficiency was estimated to be 2.0%. The results indicated the slight increase of repair efficiency by HDR because of cell-cycle dependent activation of SpyCas9. However, it is possible that the difference observed was just a consequence of the low sensitivity of this assay (~ 1–2%). In such cases, next generation sequencing (NGS), tracking of indels by decomposition (TIDE), or tracking of insertion, deletions and recombination events (TIDER)[42,43] methods could be effective options for evaluation. When HDR experiment using template plasmid DNA was performed for *H2B* gene, no HDR event could be detected. Then the phenomenon was further examined in next experiments using single-stranded donor oligonucleotides (ssODN). The ratio of off-target mutation in each condition was evaluated with the T7E1 assay (Fig. 5b). Gene editing by SpyCas9 alone showed an 8.3% off-target rate at *MYBPC2* gene, which has two-base mutations compared with *AAVS1* target sequence. Co-expression of AcrIIA4 and SpyCas9 did not show on- or off-target mutations, neither. To our surprise, no bands indicating off-target mutation by T7E1 digestion were observed in the co-expression of AcrIIA4-Cdt1 and SpyCas9. The results suggest that the co-expression of AcrIIA4-Cdt1 and SpyCas9 could be used as a method to increase HDR efficiency as well as to decrease or suppress off-target effects.

**Precision genome editing by HDR using ssODN.** It has been reported that the HDR repair process can work efficiently using

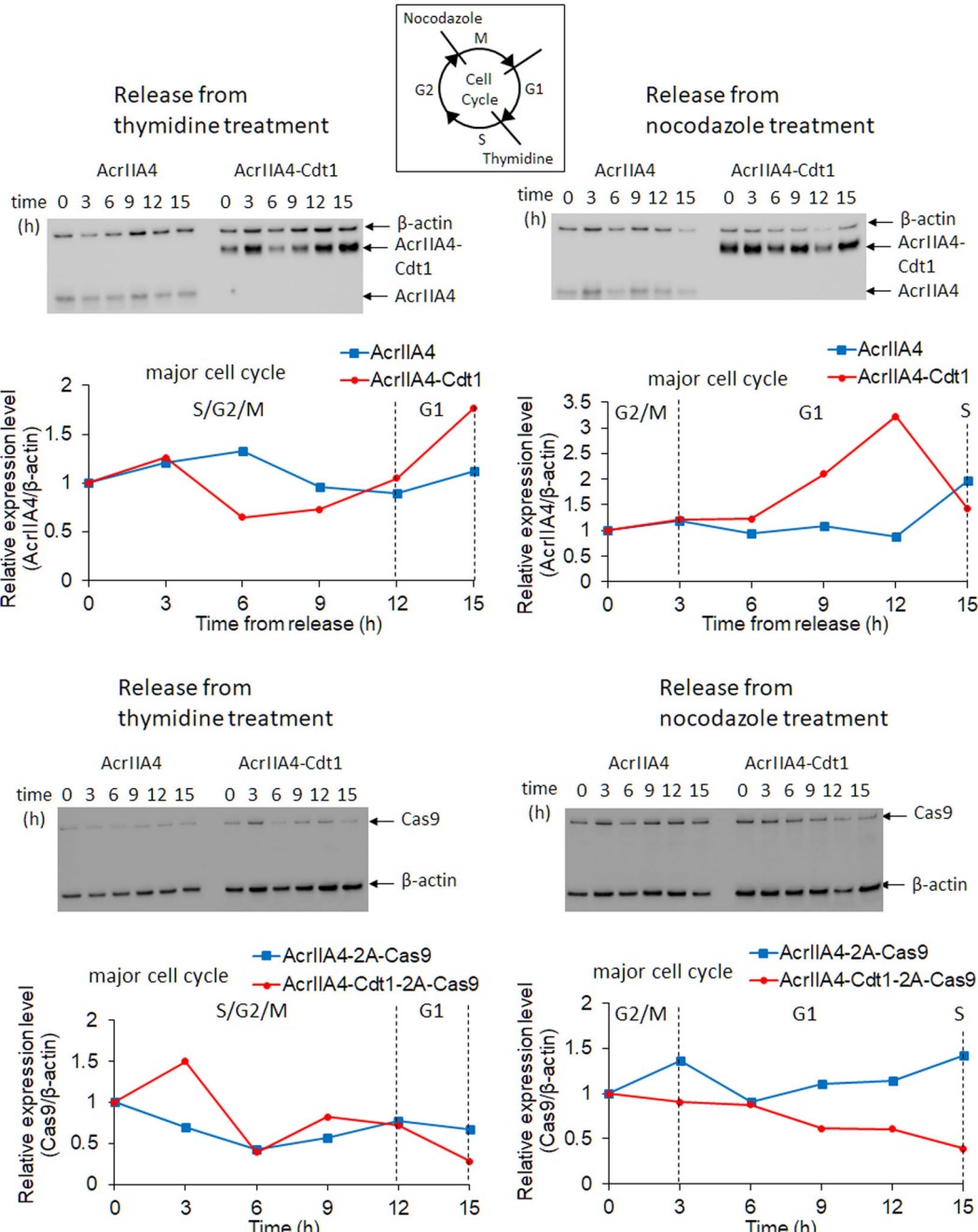

**Fig. 3 Expression analysis of anti-CRISPR or SpyCas9 expression by western blotting.** Stable cell lines which express AcrIIA4/AcrIIA4-Cdt1 and SpyCas9 were synchronized by double thymidine block or nocodazole treatment. The protein was extracted from these cells at the specified timepoints after release from the drugs. Top graphs show change of expression level of AcrIIA4/AcrIIA4-Cdt1. Bottom graphs show change of expression level of SpyCas9. Relative expression levels were calculated from the ratio of intensity between AcrIIA4/SpyCas9 and β-actin. Major cell cycle reflected the results in Supplementary Fig. 3. Actual blotting images are shown in Supplementary Fig. 4.

ssODN as template DNA[44–46]. Advantages of ssODN also include easier production, lower cost, faster reaction[47,48], and less unexpected integration, compared with double stranded DNA templates including plasmid DNA containing homology arms[49–51]. Moreover, the slight increase observed in HDR activity in Fig. 5b could be caused by inefficient cleavage of target site by using template plasmid DNA which also has two target DNA sequences. Thus, the use of ssODN was tested to increase HDR efficiency for cell-cycle dependent SpyCas9 activation (Fig. 6a). Cells were transfected with plasmids encoding SpyCas9, AcrIIA4-2A-Cas9, or AcrIIA4-Cdt1-2A-Cas9. After selection by Hygromycin,

further transfection of sgRNA coding plasmid and ssODN template DNA was performed by electroporation. Precision editing through HDR and mutation through NHEJ at on- and off-target sites were analyzed by HindIII or T7E1 digestion 72 h after the second transfection. In the analyses, three target genes, *AAVS1*, *EMX1*, and *VEGFA*, were examined preparing three sgRNA for each target site. For *AAVS1*, HDR efficiency was increased approximately by 1.7-fold when AcrIIA4-Cdt1 was co-expressed with SpyCas9 compared to that with SpyCas9 expression alone (Fig. 6b). Co-expression of AcrIIA4-Cdt1 and SpyCas9 repressed the target mutation by 79.1%. Off-target mutagenesis at *MYBPC2*

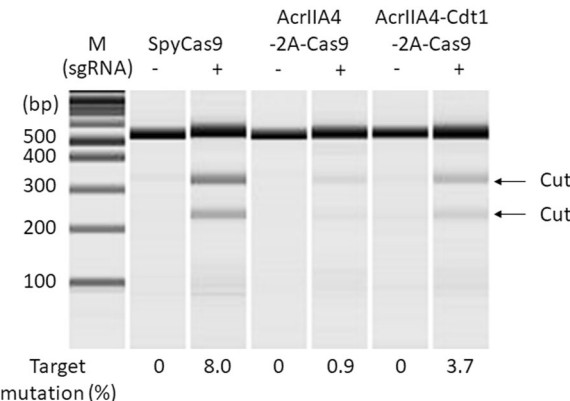

**Fig. 4 Inhibition of SpyCas9 activity by using AcrIIA4-2A-Cas9 and AcrIIA4-Cdt1-2A-Cas9 expressing vectors.** Mutation was detected by T7E1 assay and microchip electrophoresis. The editing efficiency was calculated by the formula; $100 \times (1-\sqrt{1 - (b + c)/(a + b + c)})$, where "a" is the integrated intensity of the undigested PCR product, and "b" and "c" are the integrated intensities of each cleavage product.

gene was 2.1% when SpyCas9 was expressed alone, however it was efficiently reduced to 0.4% when AcrIIA4-Cdt1 was co-expressed. Compared with the result in Fig. 5, ssODN showed more efficient HDR than plasmid DNA. This result could be caused by the different transfection efficiencies or low production efficiency of dsDNA template from plasmid by SpyCas9 cleavage. A different *EMX1* targeting sgRNA was used to confirm whether the result was similar at a different target site (Fig. 6c). The efficiency of HDR using AcrIIA4-Cdt1 was increased approximately by 4.0-fold compared to that using SpyCas9 alone. At target or off-target site 1 (*HCN1* gene), the mutation ratio was decreased by 86.5%. Moreover, the mutation ratio at off-target site 2 (*MFAP1* gene) was decreased from 8.5% to 0.6% using AcrIIA4-Cdt1. In case of *VEGFA* gene targeting, the co-expression of AcrIIA4-Cdt1 and SpyCas9 showed a 4.5-fold increase of target HDR compared with the expression of SpyCas9 only. Mutation rates at two off-target sites (*MAX* gene and non-coding site) were low (around 0.3%) when AcrIIA4-Cdt1 co-expression was used (Fig. 6d). There was no significant improvement of HDR when AcrIIA4 was used at these three target sequences. HDR with ssODN resulted on the addition of 12 bp, including the HindIII site near the target site

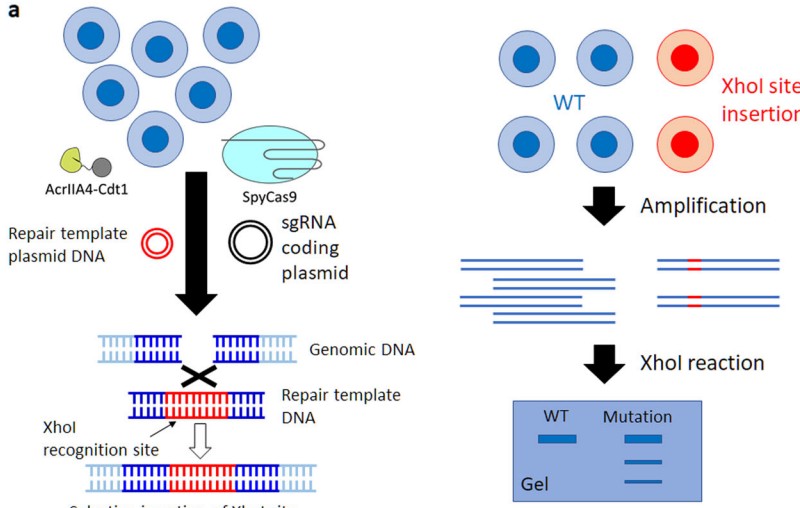

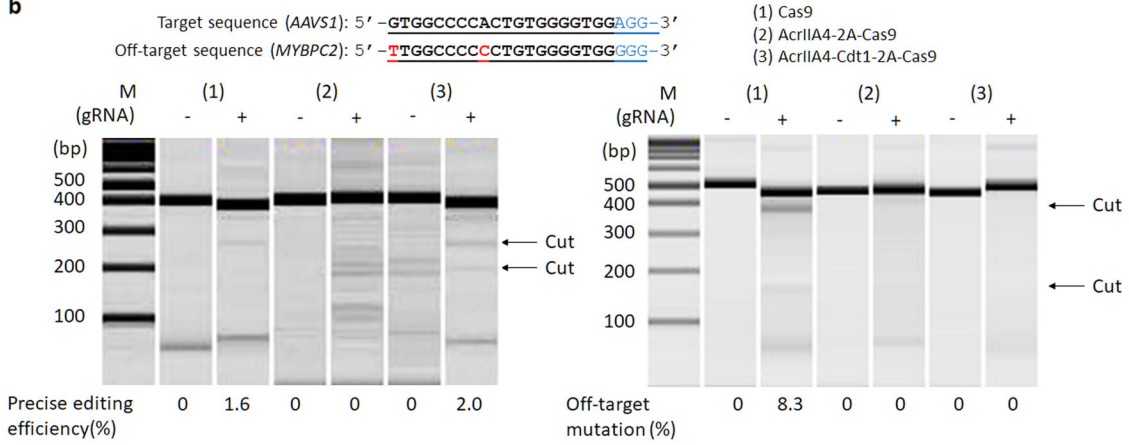

**Fig. 5 Genome editing through HDR by using repair template plasmid DNA. a** Method to confirm the HDR efficiency using XhoI restriction enzyme. **b** Results of target HDR (left) and off-target mutation (right). Each PCR product amplified from extracted genomic DNA reacted with XhoI for HDR or T7E1 for off-target mutation. The editing efficiency was calculated by the formula; $100 \times ((b + c)/(a + b + c))$ for target HDR, $100 \times (1-\sqrt{1 - (b + c)/(a + b + c)})$ for off-target mutation, where "a" is the integrated intensity of the undigested PCR product, and "b" and "c" are the integrated intensities of each cleavage product.

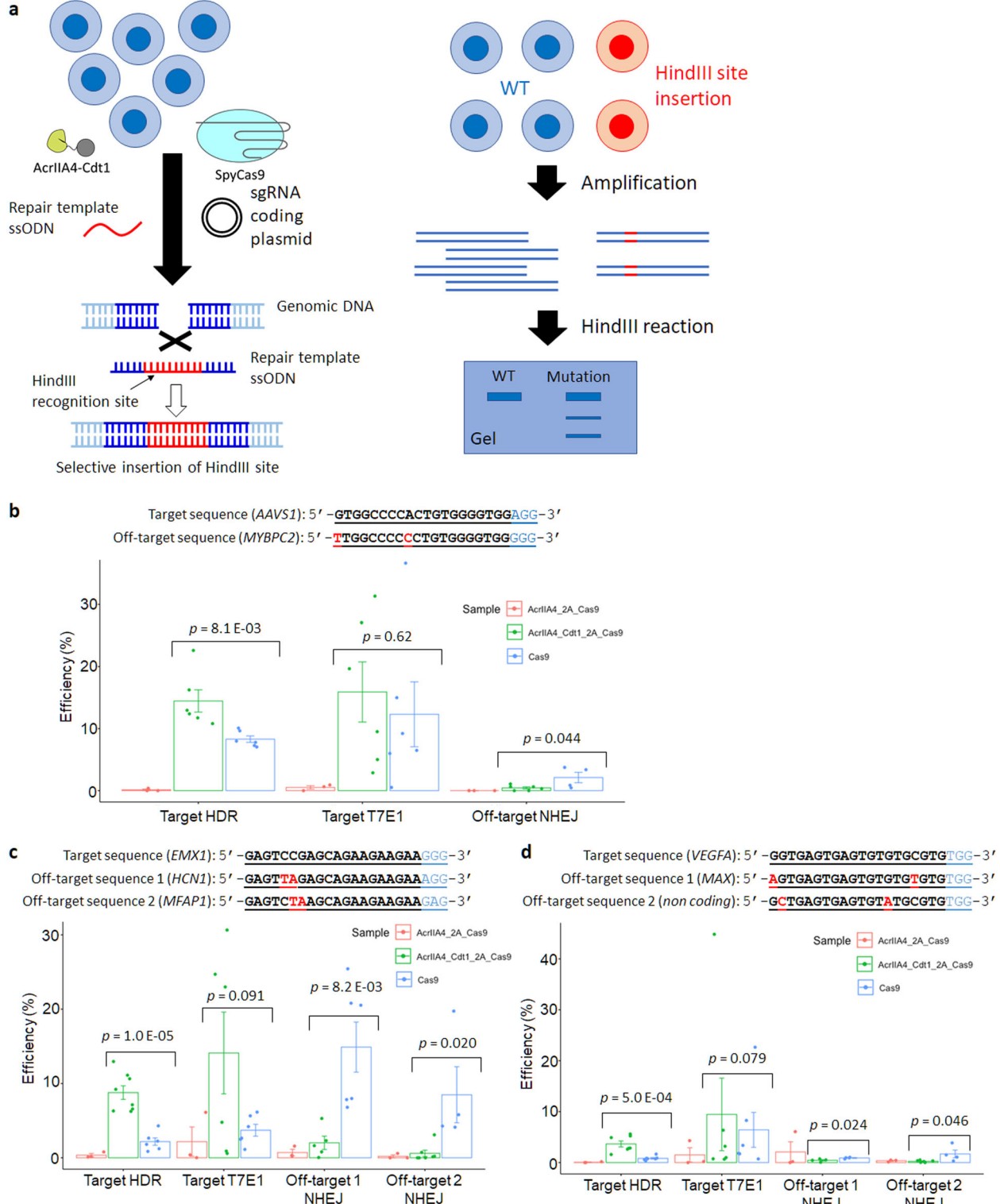

**Fig. 6 Genome editing through HDR by using repair template ssODN. a** Method to confirm the HDR efficiency using HindIII restriction enzyme. **b–d** Results of specific HDR and off-target mutation at three different targets (**b**: *AAVS1* gene, **c**: *EMX1* gene, **d**: *VEGFA* gene). Each PCR product amplified from extracted genomic DNA reacted with HindIII for target HDR or T7E1 for target T7E1 and off-target mutation. Target T7E1 includes mutagenesis by HDR and NHEJ. The editing efficiency was calculated by the formula; $100 \times ((b + c)/(a + b + c))$ for HDR, $100 \times (1-\sqrt{1 - (b + c)/(a + b + c)})$ for off-target mutation, where "a" is the integrated intensity of the undigested PCR product, and "b" and "c" are the integrated intensities of each cleavage product. $n = 6$ for SpyCas9 and AcrIIA4-Cdt1-2A-Cas9 samples. $n = 3$ for AcrIIA4-2A-Cas9 sample. Actual electrophoresis images are shown in Supplementary Fig. 6. Significance in difference was tested by Student's *t*-test.

(Supplementary Fig. 5). Target T7E1 analysis was conducted using genomic template edited with ssODN, which results in total mutation efficiency by NHEJ and HDR. To calculate the HDR/NHEJ rates, HDR efficiency by HindIII digestion was subtracted from total mutation efficiency by T7E1 digestion, resulting in NHEJ efficiency (Table 1). Editing by SpyCas9 with AcrIIA4-Cdt1 showed increased HDR/NHEJ rates in all target sites compared to that of SpyCas9 alone.

**Reduction of off-target effect by combining with truncated sgRNA.** It has been shown that the accuracy of genome editing, meaning increased HDR efficiency and reduction of off-target effects, can be achieved using SpyCas9 with AcrIIA4-Cdt1. However, about a 2% off-target mutation rate was still seen for *EMX1* gene (Fig. 6c, off-target 1). A method using truncated sgRNA[52] was applied for combined use with AcrIIA4-Cdt1 to further reduce the off-target mutation (Fig. 7a). Two new truncated sgRNAs were constructed targeting the *EMX1* and *VEGFA* genes and reduction of off-target mutations was assessed using the T7E1 assay. No off-target mutation could be observed without reducing the target HDR efficiency when truncated sgRNA targeting the *EMX1* gene was used (Fig. 7b). On the other hand, a 0.4% off-target mutation rate at

the *MAX* gene and 0.27% off-target mutation rate at the non-coding gene were detected when the truncated sgRNA targeting the *VEGFA* gene was used (Fig. 7c). It was shown that the effect of truncated sgRNA is effective but not compatible with any sequence for suppression of off-target mutation.

## Discussion

AcrIIA4-Cdt1 was constructed for precise genome editing using SpyCas9 derived from *Streptococcus pyogenes*. The target mutation rate by NHEJ was decreased when SpyCas9-sgRNA and AcrIIA4-Cdt1 were co-expressed due to SpyCas9 inhibition by AcrIIA4. A dose-dependent increase of SpyCas9 inactivation by AcrIIA4-Cdt1 was also confirmed (Fig. 3). However, it was difficult to efficiently control the activity of SpyCas9 by using different vectors having SpyCas9 or anti-CRISPR, possibly due to variable amounts of plasmids in each cell. Therefore, an episomal vector coding SpyCas9 and AcrIIA4-Cdt1 genes via self-cleaving peptide 2A was newly constructed. It was confirmed that the amount of ArIIA4-Cdt1 was dependent on cell cycle, that is, increased in $G_1$ and decreased in the $S/G_2/M$ phases, indicating that Cdt1 could be captured in the proteasome degradation in the cells. Change of SpyCas9 expression level was not evident even when simultaneously expressed with AcrIIA4-Cdt1.

AcrIIA4-Cdt1 showed efficient reduction of mutagenesis by NHEJ and off-target effects. In addition, the efficiency of HDR was increased by the use of AcrIIA4-Cdt1 with SpyCas9. These results suggest that the degradation of anti-CRISPR at the $S/G_2$ phase activates SpyCas9 and promotes DNA repair through HDR. The use of ssODN as a template enhanced HDR efficiency. When template plasmid DNA is used, a further step of SpyCas9 cleavage to make a short double stranded DNA could become a bottle neck of efficiency. Thus, it is considered that ssODN can be used more efficiently as a template in the $S/G_2$ phase in the AcrIIA4-Cdt1 and SpyCas9 co-expressing cells. In addition, the

**Table 1 HDR/NHEJ ratio for editing using ssODN.**

| HDR/NHEJ by | Target sites | | |
|---|---|---|---|
| | *AAVS* | *EMX1* | *VEGFA* |
| SpyCas9 | 2.1 | 1.4 | 0.15 |
| AcrIIA4-2A-Cas9 | 0.34 | 0.19 | 0.037 |
| AcrIIA4-Cdt1-2A-Cas9 | 10 | 1.6 | 0.63 |

NHEJ efficiency was calculated by subtracting target HDR value from target T7E1 value in Fig. 6 (b–d).

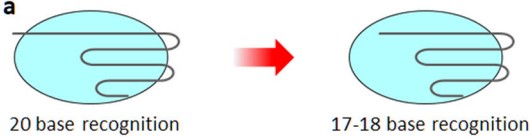

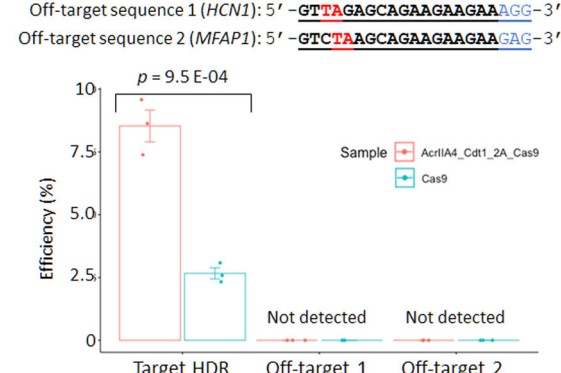

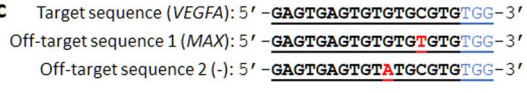

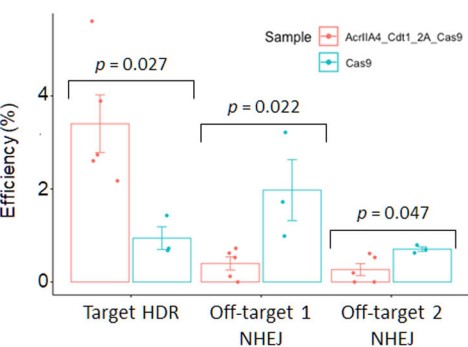

**Fig. 7 Genome editing using truncated sgRNA. a** Truncated sgRNA whose target sequence is shorter than the normal one. **b,c** Results of target HDR and off-target mutation at two different targets (**b**: *EMX1* gene, **c**: *VEGFA* gene). Each PCR product amplified from extracted genomic DNA reacted with HindIII for HDR or T7E1 for off-target mutation. The editing efficiency was calculated by the formula; 100 × ((b + c)/(a + b + c)) for HDR, 100 × (1−sqrt(1 − (b + c)/(a + b + c))) for off-target mutation, where "a" is the integrated intensity of the undigested PCR product, and "b" and "c" are the integrated intensities of each cleavage product. *n* = 3 for *EMX1* samples and *VEGFA* SpyCas9 sample. *n* = 5 for *VEGFA* AcrIIA4-Cdt1-2A-Cas9 sample. Actual electrophoresis images are shown in Supplementary Fig. 6. Significance in difference was tested by Student's t-test.

use of ssODN increase target HDR/NHEJ ratio compared with the use of SpyCas9 alone. For *VEGFA* target, HDR/NHEJ ration was less than 1, which means target NHEJ showed higher efficiency than HDR. However, increased HDR/NHEJ ratio was observed, and HDR efficiency could be increased by optimizing the homology arm of ssODN.

Our system showed a higher increase in HDR efficiency than the SpyCas9-Geminin fusion system that was previously reported[19]. Two reasons sought to be suggested. The first is the rapid recovery of active SpyCas9 from its suppressed status by anti-CRISPR. This enables SpyCas9 to cleave the target sequences promptly after cells enter the S phase. The second is the effect of SpyCas9 activity by fusing Geminin to its N-terminus. In our anti-CRISPR system, SpyCas9 is in the native form except for the addition of a proline residue at the N-terminus after T2A peptide cleavage. On the other hand, SpyCas9 is fused with Geminin (1–110), which is approximately 12 kDa, at its C-terminus, which could reduce SpyCas9 activity.

For further reduction of off-target effects, a truncated sgRNA was combined with AcrIIA4-Cdt1. The method was effective and the ratio of HDR to the off-target mutation was increased. Importantly, no off-target mutation was detected when the truncated sgRNA targeted the *EMX1* gene, in which the off-target sequence has two mismatched bases. Truncation of seed sequence of sgRNA could significantly affect the stability of SpyCas9-sgRNA-target DNA tertiary complex. Nevertheless, in case of the *VEGFA* gene, off-target effects were not completely suppressed at the *MAX* and non-coding genes. These off-target sites have single mismatched bases adjacent to G-C pairs that might be complementary to the affinity of DNA-sgRNA-SpyCas9 ternary complex. The ratio of HDR to off-target mutation in this system could be further improved by using high fidelity Cas9 or Cas9-sgRNA RNP[53–55]. Moreover, it is expected that the cell cycle dependent activatable system could be widely applied to the other combinations of CRISPR-Cas systems and anti-CRISPRs[56,57].

## Methods

**Plasmid construction**. Plasmid DNA encoding AcrIIA4 and FLAG tag genes was synthesized by Eurofins Genomics. The pFucci-G1 Orange Expression vector was purchased from MBL. The gRNA Cloning Vector and SpyCas9 were gifts from George Church (Addgene plasmids # 41824 and 41815). The AcrIIA4 fragment was produced by digesting amplified DNA with BamHI and BstXI. This fragment was ligated into the pFucci-G1 Orange expression vector at the N-terminus of hCdt1(30–120) to construct AcrIIA4-hCdt1(30–120) plasmid DNA. The AcrIIA4 fragment was amplified by CMV primer and Acr-REsite_XbaI_Rv and digested by BamHI and XbaI. This fragment was ligated into the pFucci-G1 Orange vector, which was digested by BamHI and XbaI to construct AcrIIA4 plasmid DNA. To introduce the NLS, the primers BamHI_NLS-AcrIIA4_Fw and Acr-REsite_XbaI_Rv were used. DNA fragment was amplified and digested by BamHI and XbaI, then inserted into the original vector, which was digested by BamHI and XbaI. New plasmid DNAs encoding AcrIIA4-2A-Cas9 or AcrIIA4-Cdt1-2A-Cas9 were constructed using Gibson Assembly. AcrIIA4, AcrIIA4-Cdt1, and SpyCas9 fragments were amplified by PCR. NotI-treated pEBMulti-Hyg (FUJIFILM Wako) and each fragment were inserted into the pEB vector using the Gibson Assembly Master Mix (NEB). All primer sequences are shown in Supplementary Table 1.

**Cell culture and transfection**. 293A cells (Thermo Fisher ScientificA) were maintained in DMEM supplemented with 10% fetal bovine serum (FBS) and penicillin/streptomycin at 37 °C in an atmosphere of 5% CO₂. After introduction of the episomal vector encoding Cas9, AcrIIA4-2A-Cas9, or AcrIIA4-Cdt1–2A-Cas9 by Lipofectamine 3000, the cells were selected using 350 µg/mL Hygromycin B solution (FUJIFILM Wako) for 3–7 days. Lipofectamine 3000 (Thermo Fisher Scientific) was used for western blots and assessment of plasmid amount. Neon® Transfection System 10 µL kit (Thermo Fisher Scientific) was used to assess endogenous HDR activity. In western blot analysis and time lapse observation, 500 ng of plasmid DNA were transfected by lipofection into 293A cells grown to 80–90% confluency. To examine inhibition at the target site, repair template plasmid and sgRNA plasmid (each 250 ng) were transfected into 293A cells grown to 80–90% confluency by lipofection. In the HDR assessment using ssODN as the

template, 50 pmol of ssODN and 250 ng of sgRNA plasmid were transfected into $5 \times 10^4$ cells using a pulse voltage of 1245 V, pulse width of 10 ms, and three pulses.

**T7E1 assay and restriction enzyme assay**. 293A cells were seeded on 24-well plates (Greiner Bio-One) at a density of $4 \times 10^4$ cells/well. After 24 h, cells were transfected with 500 ng of plasmid DNA. Twenty-four hours after transfection, the transfected cells were seeded in wells of a 6-well plate with DMEM containing 350 µg/mL Hygromycin B and cells were selected during incubation for one week. The transfected cells were seeded in wells of a 24-well plate at a density of $4 \times 10^4$ cells/well. Cells were incubated for 24 h and transfected with two kinds of plasmid DNA encoding sgRNA and template DNA. Forty-eight hours after transfection, genomic DNA was extracted using the QIAamp DNA Mini Kit (Qiagen). Genomic DNA (100 ng) was amplified using Herculase II Fusion DNA polymerase (Agilent) with T7E1 primers of each target. The PCR conditions used for *AAVS1* target, *EMX1* target and off-target, and vascular endothelial growth factor A (*VEGFA*) target and off-target, was 95 °C for 3 min for the first denaturation; pre-amplification using 10 cycles of 98 °C for 10 s, 72 °C to 62 °C (−1 °C per a cycle) for 20 s, and 72 °C for 30 s; amplification using 25 cycles of 98 °C for 10 s, 62 °C for 20 s, and 72 °C for 30 s; and a final extension at 72 °C for 3 min. All primer sequences are shown in Supplementary Table 1. The products were stored 4 °C in 3% dimethylsulfoxide until used. PCR conditions of other genes followed manufacturers' manuals. PCR fragment DNA was purified using the QIAquick PCR Purification Kit (Qiagen). Fragment DNA (200 ng) was annealed in 19 µL of a solution containing ed 2 µL of 10 × NEBuffer 2 (NEB) using 95 °C for 10 min, 95 °C to 25 °C at a ramp rate of 0.1 °C/s, and 4 °C. The annealed DNA received 1 µL of T7 endonuclease 1 and was incubated at 37 °C for 1 h. Reacted samples were purified by the QIAquick PCR Purification Kit (Qiagen). DNA fragments were analyzed using MultiNA (SHI-MADZU). The indel efficiency was calculated as:

$$100 \times (1 - (1 - (a+b)/(a+b+c))^{\wedge}(1/2))$$

where "*a*" and "*b*" represent the areas of cleaved fragments and "*c*" is the area of an uncleaved fragment. In the restriction enzyme assay, 200 ng of amplified DNA reacted with 0.5 µL of XhoI or HindIII (NEB) in Cutsmart buffer (NEB) and 1 × bovine serum albumin (NEB) at 37 °C for 1 h (XhoI) or 3 h (HindIII). Reacted samples were purified by ethanol precipitation. DNA fragments were analyzed using MultiNA. The indel efficiency of indel was calculated as:

$$100 \times ((a+b)/(a+b+c))$$

where "*a*" and "*b*" represent the areas of cleaved fragments and "*c*" represents the area of an uncleaved fragment.

**Time lapse observation of mKO2-Cdt1 (30–120)**. 293A cells were seeded on 24-well plates at $4 \times 10^4$ cells/well and cultured in high-glucose DMEM (FUJIFILM Wako) containing 10% FBS and penicillin/streptomycin, at 37 °C in 5% CO₂ for 24 h. 500 ng of pFucci-G1 Orange Expression vector was transfected into 293 A cells using Lipofectamine 3000 following the manufacturer's protocol. Then, transfected cells were seeded on 35 mm glass bottom dish (Greiner Bio-One) 24 h after transfection using phenol red free high-glucose DMEM (Thermo Fisher Scientific) containing 10% FBS and penicillin/streptomycin. Expression of mKO2 was observed every 1 h up to 24 h by using FLUOVIEW FV10i (Olympus).

**Immunocytochemistry**. 293A cells were seeded into 24-well plates at $4 \times 10^4$ cells/well and cultured in high-glucose DMEM containing 10% FBS and penicillin/streptomycin, at 37 °C in 5% CO₂ for 24 h. 500 ng of plasmids were transfected to cells using Lipofectamine 3000 following the manufacturer's protocol. 24 h after transfection, growth medium was changed to fresh medium, and cells were cultured for a further 24 h. Transfected cells were seeded on 35 mm glass bottom dish 24 h before observation. Cells were fixed by 4% formaldehyde solution which was diluted from 16% formaldehyde solution (Thermo Fisher Scientific) into PBS for 10 min at room temperature (rt). Cells were permeabilized by 0.1% TritonX-100 (Merck Millipore) for 10 min at r.t. and blocked by Blocking One (Nacalai tesque, Kyoto, Japan) for 1 h at rt. Then, cells were incubated with anti-FLAG tag antibody (SIGMA) for 1 h at rt and anti-Mouse IgG (H + L) Cross-Adsorbed Secondary Antibody which is labeled by Alexa Fluor 594 (Thermo Fisher Scientific) for 30 min at rt. Nucleus of cells were stained by Hoechst 33258 (DOJINDO) for 15 min at rt. Observation of stained cells was performed using FLUOVIEW FV10i.

**Statistics and reproducibility**. The data were represented as mean values ± standard error using more than three replicates. Significances in difference in Figs. 6 and 7 were tested by Student's t-test. All cell samples except for those represented in Fig. 3 were evaluated in at least biological triplicates. In vitro biochemical experiments were performed three independent times.

## Data availability
Time-lapse imaging analysis of mKO2-Cdt1 expression, cellular localization analysis of AcrIIA4 and AcrIIA4-Cdt1, cell cycle analysis by FACS, and full blotting images are available in Supplementary Information. Source data underlying plots shown in figures are provided in Supplementary Fig. 6. Source Data are available in Supplementary Data 1. All other data (if any) are available upon reasonable request.

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

## Acknowledgements

The authors would like to thank Ms. Mayumi Fujisawa and Ms. Maiko Hoshino for their technical assistance, and Editage (www.editage.com) for English language editing. This work was supported in part by the New Energy and Industrial Technique Development Organization (NEDO) of Japan, the Japan Society for the Promotion of Science (JSPS) KAKENHI (JP25410171 and JP16K01931 to W.N.), JSPS Fellows (17J08531 to D.M.) and

Grants-in-Aid for Scientific Research on Innovative Areas (JP24119506, JP26119703 (Synthetic Biology), and JP16H01420 (Resonance Bio) to W.N.).

## Author contributions

W.N. provided the scientific direction and the overall experimental design for the studies. W.N. and D.M. designed and performed all experiments. W.N., D.M. and H.T. wrote the manuscript.

## Competing interests

Tokyo Medical and Dental University and Hiroshima University have filed a patent application broadly relevant to this work. D.M., H.T. and W.N. are the investigators of record listed on the patent application.
