## [Peer Review File · Communications Biology]

Reviewers' comments:

Reviewer #1 (Remarks to the Author):

Matsumoto et al., reports a system in which a cell-cycle-dependent degron regulated anti-CRISPR protein is used to achieve efficient Cas9-mediated homology-directed repair (HDR). To this end, authors have taken advantage of Cdt1 and hGeminin degrons which have been used in Fucci system to visualize the cell cycle progression. These degrons can be fused to other proteins to have reciprocal oscillating levels throughout the cell cycle: hGeminin fusion degrades in G1 and Cdt1 fusion degrades in S/M/G2. hGeminin previously has been fused to SpyCas9 so that the protein is expressed only in S/M/G2 when HDR is most active. In this study, authors sought out to use AcrIIA4 with Cdt1 which will be active in G1 and degrade in S/M/G2, thus inhibiting SpyCas9 when NHEJ is dominant while allowing gene editing in HDR-active phases.

Overall, the study has its creativity and value in establishing a cell-autonomous system without the use of small molecule inhibitors of NHEJ or cell synchronization methods, however, current manuscript lacks the depth of investigation. Additional experiments addressing major concerns (see below) that provide a more thorough study will be required for publication of this manuscript.

Major points:

For estimating the efficiency for indels and precise editing authors have used T7E1 and restriction enzyme digestion, respectively. Two methods have been outdated in the field mainly due to low sensitivity of these assays. The sensitivity of the assays ($\sim 1-2\%$) is problematic especially the data presented in the manuscript have overall low efficiencies (generally $< 10\%$). Comparing indels that are below the detection limit of these assays can be very inaccurate. I would highly encourage authors to evaluate efficiencies by at least Sanger sequencing (if NGS is not an option). PCR products can be directly used for Sanger sequencing for both indels and HDR events and analyzed using existing software tools such as TIDE or TIDER

(<http://nar.oxfordjournals.org/content/42/22/e168>;

<https://academic.oup.com/nar/article/46/10/e58/4925757>)

Although NHEJ is reduced with AcrIIA4-Cdt1 when no donor was provided (Figure S5), it is not clear what are the levels of NHEJ events in the presence of donor (specifically, what is the ratio of HDR to NHEJ?). Authors have reported only precision gene editing for on-target sites. What are the NHEJ-mediated indels (non-HDR events) at on-target sites? A true indicator of "precision gene editing" is not really comparing editing at on- vs off-target site, but rather comparing indels vs HDR for the on-target site to evaluate how effective AcrIIA4-Cdt1 is in suppressing NHEJ-mediated, imprecise editing while allowing precise HDR integration of a donor template. To this end, it would be fair to compare the indels vs HDR events for on-target sites as well.

Instead of using episomal vectors for Cas9 and AcrIIA4 expression, would it make more sense to have an a single-copy integration of the cassette in cells to have endogenous level of expression of AcrIIA4-Cdt1 and SpyCas9? Presumably the protein levels could be overwhelming for ubiquitin-degradation pathway. Especially there seems to be incomplete degradation of AcrIIA4-Cdt1 (Figure S4) and its residual inhibitory function may reduce the overall gene editing by Cas9. Although the protein levels vary depending on the cell cycle stages (Figure S4), there is still significant amount of residual AcrIIA4-Cdt1 expression in S/G2/M (only ~ 0.5 reduction), which may be enough to inhibit SpyCas9 gene editing.

AcrIIA4-Cdt1 seems to increase HDR (maximum ~ 4 -fold compared to SpyCas9 alone control). Provided that SpyCas9-hGeminin previously has been shown to increase the HDR with a relatively low fold improvement, why authors have not tried combining both SpyCas9-hGeminin and AcrIIA4-Cdt1? Either additive or synergistic effects of both systems could be achieved.

Other minor points:

Has author tried N- vs C-terminus fusion of Cdt1 to AcrIIA4? Inhibitory potential and degradation

of AcrIIA4 could be affected by different terminal fusion.

Regarding Figure S4: Did authors check the efficiency of T2A cleavage of the AcrIIA4-Cdt1-T2A-SpyCas9 from their western blot experiments? For the western blot experiments, how reproducible is the data? Is the blot representative of how many biological replicates?

Regarding Figure S5: Was the genomic DNA collected from a heterogeneous population of cells undergoing different cell cycle stages? The authors suggest that "cell-cycle dependent expression of AcrIIA4-Cdt1 could limit the activity of SpCas9 during G1 phase even if the inhibition is weaker than when AcrIIA4 is expressed alone." Is AcrIIA4-Cdt1 not as effective as wildtype AcrIIA4 for inhibiting SpyCas9 editing generally? One could argue that 3.7% editing could have been as a result of NHEJ-mediated indels in cell cycle phases outside of G1 (as NHEJ pathway can take place throughout). Is there a possibility that reduced editing by AcrIIA4-Cdt1 is due to Cas9 activity in S/M/G2, and not due to ineffective inhibition of editing in G1?

Regarding Figure 3: I recommend authors to plot each data point (n=3 or 6). Especially, a bar graph AcrIIA4-T2A-Cas9 in Figure 3D has a very large variation.

Since suppression of off-target NHEJ-mediated indels by anti-CRISPR proteins have been previously reported, the data shown in this manuscript in which both AcrIIA4 and AcrIIA4-Cdt1 repressing off-target editing is not at all surprising. Please take a look at this manuscript: Aschenbrenner et al., 2020 (<https://advances.sciencemag.org/content/6/6/eaay0187>)

Authors need to read the entire manuscript to correct any typos and incomplete sentences to convey the messages clearly. Here are some that caught my eyes:

- In introduction: "Although the CRISPR technology is the most useful method for genome editing, off-target effects that cause mutations at pseudo target DNA sequences could occur as other editing technologies." The latter part of sentence is somewhat vague.
- In introduction: "As the repaired sequence traces template DNA, editing by HR is precision." Please rephrase.
- In result: "In case targeting VEGFA gene. target precise editing occurred 4.5-fold more frequently when AcrIIA4-Cdt1 was expressed with SpCas9 than when only SpCas9 was expressed." Change "." to ","
- Figure S2 legend typo: Change "Conformation" to "Confirmation"

Reviewer #2 (Remarks to the Author):

Matsumoto et al. CommsBio

This is a very intriguing paper from Matsumoto et al. that show an application for AcrIIA4 anti-CRISPR protein when fused to Cdt1, which gets the Acr degraded in S/G2 phases of the cell cycle. Most interestingly, the authors show something I haven't seen before, an increase in on-target HR editing due to an anti-CRISPR inhibiting Cas9 during cell cycle phases that do not favor HDR. I think the idea is very clever, and is clearly superior to other approaches that rely on synchronizing cells, small molecule drugs, or Cas9 fusions. I have a number of issues with the manuscript's presentation however and I have made those specific comments below. However, I would be happy to re-consider the paper when these changes (mostly textual) are made, to make the paper strong, in my opinion.

A consistent issue is the choice to use model cartoons as main figures and to put real (and important) data in the supplement. While the cartoons can be helpful, they are not always needed and the data (in many cases, specified below) should be shown in the main paper.

Sometimes the authors are unclear if referring to real data or hypothetical:

e.g. "Consequently, when Cas9 was co-expressed with AcrIIA4-Cdt1 in the cells, SpCas9 activity is inhibited during the G1 phase, and when AcrIIA4-Cdt1 was degraded in the S/G2 phases SpCas9 activity is regained." This is upstream of a reference to fig.1 which is just a model hypothetical. Consider rephrasing to be more clear about what is real data.

On that note, I would suggest that Figure S3 is a main figure as this is an important control that should be shown. Figure S2 could also be included. While I find the model in Fig 1 helpful, there is no reason to not include the control experiments as real figures.

I have the same suggestion for Figure S4B, where the expression of Cas9 and AcrIIA4 were analyzed by western blotting. This is very important to show. Figure S5 should also be in the main text.

Does Cdt1 fusion increase the maximum level of AcrIIA4? Does it have a stabilizing effect?

In looking at Fig S4B, while I see and agree with the change in AcrIIA4 levels when fused to Cdt1, the impact is fairly subtle, 2-3 fold. The authors should be clear about the fold change in the text and also give the reader some indication if that is the expected change based on what is known about cdt1 or what can be seen with the control fluorescence experiments.

Figure S5 legend: a "newly constructed vector" is not helpful. Should just state what it is, likewise in the associated text. This happens again with Fig 2A associated text. That it is new, is not informative, please describe it specifically.

The text associated with Fig S5 concludes that the decrease of NHEJ suggested that AcrIIA4-Cdt1 could be controlled by cell cycle. This is unclear. Are the authors comparing the 0.9% to 3.7% and saying that the fusion is limiting the activity of AcrIIA4 or comparing 8% to 3.7% saying that the decreased NHEJ shows AcrIIA4 is still working? This is unclear and somewhat misleading. If the latter, this conclusion is not warranted as the decreased NHEJ could just be due to AcrIIA4 activity being weakened by a fusion.

The experiment where precise editing of 1.6% increases to 2.0% should refer to Figure 2? Has significance testing been applied here? While the authors describe this as a slight increase (accurate), they conclude that this is "because of cell-cycle dependent activation of Cas9". This claim is unwarranted as it is not associated with any proof that this is not assay variability or other factors. However, the maintained on-target with no off-target is very exciting. However, the authors should declare the limit of detection for the T7E1 assay.

The changes in on-target HDR integration compared to the off-target are impressive. Are the authors comparing the % of on-target loci that have the HDR event to those that don't? are they still getting NHEJ in-dels? This should be clarified. And for off-target, this does not involve HDR, but is just an in-del mutation at a known Off target site, correct?

Given the impressive results I would suggest that the authors show representative gels from Fig 3 and 4 in the supplement.

Pertaining to fig 4, I don't actually see where in the text the authors discuss/refer to the actual AcrIIA4-Ctd1 data in Fig 4B/4C

Discussion:

Second sentence of the discussion again mentioned the NHEJ decrease when AcrIIA4-Ctd1 was used and while this may be true, isn't it the case that this result alone doesn't prove it is due to Ctd1, since the AcrIIA4 protein alone also shows lower "NHEJ rate" (due to Cas9 inhibition).

Third sentence mentions dose-dependent inhibition. Where was this shown? Would be helpful to refer to the specific figures, from the discussion.

It would be helpful if the authors compare the HR rates they are getting (via plasmid or ssODN), during uninhibited Cas9 editing are comparable to other studies, especially if same sites have been used.

Minor:

Adding line numbers would help the review process.

Abstract; "these exhibited lower editing efficiency"... is this always true? insert "often"? or "sometimes"?

Suggest "here, the anti-crispr protein..." Can make it more clear where this work starts.

Typo "showed not only increased the frequency.."

If 'precise editing' means HDR events, this should be articulated, to distinguish from off-target edits or off target HDR

Nice intro. Touches on the important points I expected to see
KD and KD both used. Please standardize. And I suggest you use ApoSpCas9 instead of just SpCas9 to distinguish from sgRNA-bound form.

Again, towards end of the intro, the authors should consider inserting a "here, we show" to distinguish between what is previous work, what would be hypothetically useful and then what they did.

Results

AcrIIA4-Cdt1(30/120) nomenclature seems odd to me

AcrIIA4-Cdt130-120?

Communications Biology manuscript**Matsumoto *et al.*, “A Cell Cycle-dependent CRISPR-Cas9 Activation System Based on an Anti-CRISPR Protein Increased the Accuracy of Genome Editing”**

Matsumoto *et al.*, reports a system in which a cell-cycle-dependent degron regulated anti-CRISPR protein is used to achieve efficient Cas9-mediated homology-directed repair (HDR). To this end, authors have taken advantage of Cdt1 and hGeminin degrons which have been used in Fucci system to visualize the cell cycle progression. These degrons can be fused to other proteins to have reciprocal oscillating levels throughout the cell cycle: hGeminin fusion degrades in G1 and Cdt1 fusion degrades in S/M/G2. hGeminin previously has been fused to SpyCas9 so that the protein is expressed only in S/M/G2 when HDR is most active. In this study, authors sought out to use AcrIIA4 with Cdt1 which will be active in G1 and degrade in S/M/G2, thus inhibiting SpyCas9 when NHEJ is dominant while allowing gene editing in HDR-active phases.

Overall, the study has its creativity and value in establishing a cell-autonomous system without the use of small molecule inhibitors of NHEJ or cell synchronization methods, however, current manuscript lacks the depth of investigation. Additional experiments addressing major concerns (see below) that provide a more thorough study will be required for publication of this manuscript.

Major points:

For estimating the efficiency for indels and precise editing authors have used T7E1 and restriction enzyme digestion, respectively. Two methods have been outdated in the field mainly due to low sensitivity of these assays. The sensitivity of the assays (~1-2%) is problematic especially the data presented in the manuscript have overall low efficiencies (generally <10%). Comparing indels that are below the detection limit of these assays can be very inaccurate. I would highly encourage authors to evaluate efficiencies by at least Sanger sequencing (if NGS is not an option). PCR products can be directly used for Sanger sequencing for both indels and HDR events and analyzed using existing software tools such as TIDE or TIDER

(<http://nar.oxfordjournals.org/content/42/22/e168>;
<https://academic.oup.com/nar/article/46/10/e58/4925757>)

Although NHEJ is reduced with AcrIIA4-Cdt1 when no donor was provided (Figure S5), it is not clear what are the levels of NHEJ events in the presence of donor (specifically, what is the ratio of HDR to NHEJ?). Authors have reported only precision gene editing for on-target sites. What are the NHEJ-mediated indels (non-HDR events) at on-target sites? A true indicator of “precision gene editing” is not really comparing editing at on- vs off-target site, but rather comparing indels vs HDR for the on-target site to evaluate how effective AcrIIA4-Cdt1 is in suppressing NHEJ-mediated, imprecise editing while allowing precise HDR integration of a donor template. To this end, it would be fair to compare the indels vs HDR events for on-target sites as well.

Instead of using episomal vectors for Cas9 and AcrIIA4 expression, would it make more sense to have an a single-copy integration of the cassette in cells to have endogenous level of expression of AcrIIA4-Cdt1 and SpyCas9? Presumably the protein levels could be overwhelming for ubiquitin-degradation pathway. Especially there seems to be incomplete degradation of AcrIIA4-Cdt1 (Figure S4) and its residual inhibitory function may reduce the overall gene editing by Cas9. Although the protein levels vary depending on the cell cycle stages (Figure S4), there is still significant amount of residual AcrIIA4-Cdt1 expression in S/G2/M (only ~0.5 reduction), which may be enough to inhibit SpyCas9 gene editing.

AcrIIA4-Cdt1 seems to increase HDR (maximum ~ 4-fold compared to SpyCas9 alone control). Provided that SpyCas9-hGeminin previously has been shown to increase the HDR with a relatively

low fold improvement, why authors have not tried combining both SpyCas9-hGeminin and AcrIIA4-Cdt1? Either additive or synergistic effects of both systems could be achieved.

Other minor points:

Has author tried N- vs C-terminus fusion of Cdt1 to AcrIIA4? Inhibitory potential and degradation of AcrIIA4 could be affected by different terminal fusion.

Regarding Figure S4: Did authors check the efficiency of T2A cleavage of the AcrIIA4-Cdt1-T2A-SpyCas9 from their western blot experiments? For the western blot experiments, how reproducible is the data? Is the blot representative of how many biological replicates?

Regarding Figure S5: Was the genomic DNA collected from a heterogeneous population of cells undergoing different cell cycle stages? The authors suggest that “cell-cycle dependent expression of AcrIIA4-Cdt1 could limit the activity of SpCas9 during G1 phase even if the inhibition is weaker than when AcrIIA4 is expressed alone.” Is AcrIIA4-Cdt1 not as effective as wildtype AcrIIA4 for inhibiting SpyCas9 editing generally? One could argue that 3.7% editing could have been as a result of NHEJ-mediated indels in cell cycle phases outside of G1 (as NHEJ pathway can take place throughout). Is there a possibility that reduced editing by AcrIIA4-Cdt1 is due to Cas9 activity in S/M/G2, and not due to ineffective inhibition of editing in G1?

Regarding Figure 3: I recommend authors to plot each data point (n=3 or 6). Especially, a bar graph AcrIIA4-T2A-Cas9 in Figure 3D has a very large variation.

Since suppression of off-target NHEJ-mediated indels by anti-CRISPR proteins have been previously reported, the data shown in this manuscript in which both AcrIIA4 and AcrIIA4-Cdt1 repressing off-target editing is not at all surprising. Please take a look at this manuscript: Aschenbrenner et al., 2020 (<https://advances.sciencemag.org/content/6/6/eaay0187>)

Authors need to read the entire manuscript to correct any typos and incomplete sentences to convey the messages clearly. Here are some that caught my eyes:

- In introduction: “Although the CRISPR technology is the most useful method for genome editing, off-target effects that cause mutations at pseudo target DNA sequences could occur as other editing technologies.” → Highlighted region is somewhat vague.
- In introduction: “As the repaired sequence traces template DNA, editing by HR is precision.” → Rephrase.
- In result: “In case targeting VEGFA gene. target precise editing occurred 4.5-fold more frequently when AcrIIA4-Cdt1 was expressed with SpCas9 than when only SpCas9 was expressed.” → Change “.” to “,”
- Figure S2 legend typo: Conformation → Confirmation

According to useful and important comments of the reviewers, we have appropriately prepared the **REVISED** manuscript. Correspondences to comments of reviewers 1 and 2 are described below point-by-point.

Comments of Reviewer #1:

Matsumoto et al., reports a system in which a cell-cycle-dependent degron regulated anti-CRISPR protein is used to achieve efficient Cas9-mediated homology-directed repair (HDR). To this end, authors have taken advantage of Cdt1 and hGeminin degrons which have been used in Fucci system to visualize the cell cycle progression. These degrons can be fused to other proteins to have reciprocal oscillating levels throughout the cell cycle: hGeminin fusion degrades in G1 and Cdt1 fusion degrades in S/M/G2. hGeminin previously has been fused to SpyCas9 so that the protein is expressed only in S/M/G2 when HDR is most active. In this study, authors sought out to use AcrIIA4 with Cdt1 which will be active in G1 and degrade in S/M/G2, thus inhibiting SpyCas9 when NHEJ is dominant while allowing gene editing in HDR-active phases.

Overall, the study has its creativity and value in establishing a cell-autonomous system without the use of small molecule inhibitors of NHEJ or cell synchronization methods, however, current manuscript lacks the depth of investigation. Additional experiments addressing major concerns (see below) that provide a more thorough study will be required for publication of this manuscript.

Major points:

For estimating the efficiency for indels and precise editing authors have used T7E1 and restriction enzyme digestion, respectively. Two methods have been outdated in the field mainly due to low sensitivity of these assays. The sensitivity of the assays (~1-2%) is problematic especially the data presented in the manuscript have overall low efficiencies (generally <10%). Comparing indels that are below the detection limit of these assays can be very inaccurate. I would highly encourage authors to evaluate efficiencies by at least Sanger sequencing (if NGS is not an option). PCR products can be directly used for Sanger sequencing for both indels and HDR events and analyzed using existing software tools such as TIDE or TIDER

(<http://nar.oxfordjournals.org/content/42/22/e168>;<https://academic.oup.com/nar/article/46/10/e58/4925757>)

Thank you for instructive suggestion. As far as we analyze the efficiency with the same method, we think relative changes of efficiency are valid. In addition, the HDR method using plasmid donor was not effective, then we changed to the method using ssODN donor, which showed increased HDR rates. However, we have added sentences to mention the detection limit of T7E1 assay and effective methods for evaluation of editing efficiency with citation of the recommended articles (page 8, lines

9-13).

Although NHEJ is reduced with AcrIIA4-Cdt1 when no donor was provided (Figure S5), it is not clear what are the levels of NHEJ events in the presence of donor (specifically, what is the ratio of HDR to NHEJ?). Authors have reported only precision gene editing for on-target sites. What are the NHEJ-mediated indels (non-HDR events) at on-target sites? A true indicator of “precision gene editing” is not really comparing editing at on- vs off-target site, but rather comparing indels vs HDR for the on-target site to evaluate how effective AcrIIA4-Cdt1 is in suppressing NHEJ-mediated, imprecise editing while allowing precise HDR integration of a donor template. To this end, it would be fair to compare the indels vs HDR events for on-target sites as well.

Thank you for your constructive opinion. We have checked target NHEJ by T7E1 assay, which also detects mutation by HDR as shown in Figure 6. We further calculated NHEJ rates by subtracting HDR value from total T7E1 value, then Table 1, which shows HDR/NHEJ ratio, was newly added. The values show the increase of HDR/NHEJ ratio by using AcrIIA4-Cdt1-2A-Cas9 compared to using SpyCas9 alone. The description of the analysis was added to the main text (p10, lines 19-24) in addition to Figure S4, which shows resulting sequence by HDR, and Table 1.

Instead of using episomal vectors for Cas9 and AcrIIA4 expression, would it make more sense to have an a single-copy integration of the cassette in cells to have endogenous level of expression of AcrIIA4-Cdt1 and SpyCas9? Presumably the protein levels could be overwhelming for ubiquitin-degradation pathway. Especially there seems to be incomplete degradation of AcrIIA4-Cdt1 (Figure S4) and its residual inhibitory function may reduce the overall gene editing by Cas9. Although the protein levels vary depending on the cell cycle stages (Figure S4), there is still significant amount of residual AcrIIA4-Cdt1 expression in S/G2/M (only ~0.5 reduction), which may be enough to inhibit SpyCas9 gene editing.

We agree that expression level of anti-CRISPR and Cas9 is important for results. Use of a single copy insertion could be a great option. However, locus for single insertion using such as Flp-in system may limit the use in various cell-types and stable expression by viral integration cannot control inserted copy numbers. We agree that the expression levels may not be optimum using episomal vector, however, the results indicate that system can work. For incomplete degradation of AcrIIA4, the time course of cell collection was every 3 hours. As indicated in Figure S1, mKO2-Cdt1 expression was completely suppressed when observed in the time course of every 1 hour. Thus, it is possible that the timing of cell collection was not the best to detect degradation of AcrIIA4-Cdt1. However, change of expression level of AcrIIA4-Cdt1 was successfully observed.

The change of the expression level was 3-fold, which we added the sentence to the main text as suggested by reviewer #2.

AcrIIA4-Cdt1 seems to increase HDR (maximum ~ 4-fold compared to SpyCas9 alone control). Provided that SpyCas9-hGeminin previously has been shown to increase the HDR with a relatively low fold improvement, why authors have not tried combining both SpyCas9-hGeminin and AcrIIA4-Cdt1? Either additive or synergistic effects of both systems could be achieved.

Thank you for very useful suggestion. Combination of SpyCas9-hGeminin and AcrIIA4-Cdt1 might be a better option. We are currently trying to fine-tune expression timing of AcrIIA4 using different cell cycle dependent proteins.

A possible reason why SpyCas9-hGeminin does not work well is that degradation of Cas9 doesn't work well because of the size of the protein. The small size of AcrIIA4 could be an advantage. In that case, it may be possible that the combination of SpyCas9-hGeminin and AcrIIA4-Cdt1 doesn't show significant improvement. There should be countless options for improvement, however, the authors believe that the data in this study is enough to show the significance of improvement of HDR rates and suppression of off-target effects.

Other minor points: Has author tried N- vs C-terminus fusion of Cdt1 to AcrIIA4? Inhibitory potential and degradation of AcrIIA4 could be affected by different terminal fusion.

Comparison between C- and N-terminus fusions have not been done because the distance of these terminus is very close according to the structure as reported (Kim, I., et. al. *Sci. Rep.* 2018, <https://www.nature.com/articles/s41598-018-22177-0.pdf>). This would lead to similar outcome.

Regarding Figure S4: Did authors check the efficiency of T2A cleavage of the AcrIIA4-Cdt1-T2A-SpyCas9 from their western blot experiments? For the western blot experiments, how reproducible is the data? Is the blot representative of how many biological replicates?

We didn't check T2A cleavage as the sequence is commonly used. The data is the single trial from each data point.

Figure S4 was changed to Figure 3 by a suggestion from reviewer #2.

Regarding Figure S5: Was the genomic DNA collected from a heterogeneous population of cells undergoing different cell cycle stages? The authors suggest that "cell-cycle dependent expression

of AcrIIA4-Cdt1 could limit the activity of SpCas9 during G1 phase even if the inhibition is weaker than when AcrIIA4 is expressed alone.” Is AcrIIA4-Cdt1 not as effective as wildtype AcrIIA4 for inhibiting SpyCas9 editing generally? One could argue that 3.7% editing could have been as a result of NHEJ-mediated indels in cell cycle phases outside of G1 (as NHEJ pathway can take place throughout). Is there a possibility that reduced editing by AcrIIA4-Cdt1 is due to Cas9 activity in S/M/G2, and not due to ineffective inhibition of editing in G1?

Figure S5 was changed to Figure 4 by a suggestion from reviewer #2. The experiment was performed without synchronizing cell cycle for Figure 4. Authors think that the sentence caused misleading. Thus, the sentences were changed as follows.

“Decrease of mutagenesis rates by NHEJ were exhibited when using vectors those have AcrIIA4-2A-Cas9 (0.9%) and AcrIIA4-Cdt1-2A-Cas9 (3.7%) compared to that of when using SpyCas9 alone (8.0%). The difference in decreased NHEJ rates between AcrIIA4 and AcrIIA4-Cdt1 suggests that the fusion is limiting the activity of AcrIIA4 by the cell-cycle dependent degradation of Cdt1. Expression of AcrIIA4-Cdt1 fusion showed higher mutagenesis rate compared to that of AcrIIA4 alone. It suggests that cell-cycle dependent expression of AcrIIA4-Cdt1 could inhibit SpyCas9 in the G1 phase and release it in the other phases while AcrIIA4 continuously inhibits SpyCas9” (p7, line 15-22).

Regarding Figure 3: I recommend authors to plot each data point (n=3 or 6). Especially, a bar graph AcrIIA4-T2A-Cas9 in Figure 3D has a very large variation.

Thank you for your suggestion. Figure 3 was changed to Figure 6. We made changes for graphs to put data plots.

Since suppression of off-target NHEJ-mediated indels by anti-CRISPR proteins have been previously reported, the data shown in this manuscript in which both AcrIIA4 and AcrIIA4-Cdt1 repressing off-target editing is not at all surprising. Please take a look at this manuscript: Aschenbrenner et al., 2020 (<https://advances.sciencemag.org/content/6/6/eaay0187>)

Thank you for your suggestion of an interesting paper. Their work is very impressive. We aimed to develop the method to increase HDR in this study, and found combined effects both increased HDR and suppressed off-target editing by using cell cycle dependent expression of anti-CRISPR. If only suppressing off-target effects by using Anti-CRISPR, we agree that it is not surprising or a new finding.

Authors need to read the entire manuscript to correct any typos and incomplete sentences to convey the messages clearly. Here are some that caught my eyes:

- In introduction: “Although the CRISPR technology is the most useful method for genome editing, off-target effects that cause mutations at pseudo target DNA sequences could occur as other editing technologies.” The latter part of sentence is somewhat vague.*
- In introduction: “As the repaired sequence traces template DNA, editing by HR is precision.” Please rephrase.*
- In result: “In case targeting VEGFA gene. target precise editing occurred 4.5-fold more frequently when AcrIIA4-Cdt1 was expressed with SpCas9 than when only SpCas9 was expressed.” Change “.” to “,”*
- Figure S2 legend typo: Change "Conformation" to "Confirmation"*

Thank you for your correction. We have made all changes as pointed out. The sentence “As the repaired sequence traces template DNA, editing by HR is precision” was changed to “In the HDR events, repair of target sequences trace template DNA” (page 2, lines 13-14).

Comments of Reviewer #2:

Matsumoto et al. CommsBio

This is a very intriguing paper from Matsumoto et al. that show an application for AcrIIA4 anti-CRISPR protein when fused to Cdt1, which gets the Acr degraded in S/G2 phases of the cell cycle. Most interestingly, the authors show something I haven't seen before, an increase in on-target HR editing due to an anti-CRISPR inhibiting Cas9 during cell cycle phases that do not favor HDR. I think the idea is very clever, and is clearly superior to other approaches that rely on synchronizing cells, small molecule drugs, or Cas9 fusions. I have a number of issues with the manuscript's presentation however and I have made those specific comments below. However, I would be happy to re-consider the paper when these changes (mostly textual) are made, to make the paper strong, in my opinion.

A consistent issue is the choice to use model cartoons as main figures and to put real (and important) data in the supplement. While the cartoons can be helpful, they are not always needed and the data (in many cases, specified below) should be shown in the main paper.

Sometimes the authors are unclear if referring to real data or hypothetical: e.g. “Consequently, when Cas9 was co-expressed with AcrIIA4-Cdt1 in the cells, SpCas9 activity is inhibited during the G1 phase, and when AcrIIA4-Cdt1 was degraded in the S/G2 phases SpCas9 activity is regained.” This is upstream of a reference to fig.1 which is just a model hypothetical. Consider

rephrasing to be more clear about what is real data. On that note, I would suggest that Figure S3 is a main figure as this is an important control that should be shown. Figure S2 could also be included. While I find the model in Fig 1 helpful, there is no reason to not include the control experiments as real figures.

I have the same suggestion for Figure S4B, where the expression of Cas9 and AcrIIA4 were analyzed by western blotting. This is very important to show. Figure S5 should also be in the main text.

Thank you for your advices. We have made changes for sentences describing model hypothetical to make it more clear as follows (page 5, lines 7-9, and page 6, lines 17-18).

“We hypothesized that when Cas9 is co-expressed with AcrIIA4-Cdt1 in the cells, SpyCas9 activity is inhibited during the G1 phase, and is regained when AcrIIA4-Cdt1 is degraded in the S/G2 phases.”

“We hypothesized that the amounts of AcrIIA4-Cdt1 and SpyCas9 could be strictly regulated as in the previous reports using T2A peptide (36-39).”

In addition, we put Figures S3, S4B, S5 to the main text as Figures 2, 3, and 4, respectively. Figure S2 shows confirmation of nuclear translocation of AcrIIA4 and AcrIIA4-Cdt1. Then it was remained in the supplementary.

Does Cdt1 fusion increase the maximum level of AcrIIA4? Does it have a stabilizing effect?

In looking at Fig S4B, while I see and agree with the change in AcrIIA4 levels when fused to Cdt1, the impact is fairly subtle, 2-3 fold. The authors should be clear about the fold change in the text and also give the reader some indication if that is the expected change based on what is known about cdt1 or what can be seen with the control fluorescence experiments.

Figure S5 legend: a “newly constructed vector” is not helpful. Should just state what it is, likewise in the associated text. This happens again with Fig 2A associated text. That it is new, is not informative, please describe it specifically.

For the increased expression level of AcrIIA4-Cdt1 compared with AcrIIA4, it might be possible that Cdt1 stabilizes AcrIIA4. We have changed the sentences as follows to mention about the fold change of expression level and relation with the result of mKO2-Cdt1 (page 7, lines 10-13).

“Decreased expression of AcrIIA4-Cdt1 fusion was observed at the S/G₂/M phase, while expression was increased at the G₁ phase. The change of expression level was about 3-fold. The results confirm that change of expression level as also observed in the mKO2-Cdt1 (Fucci) expression depends on the cell cycle (Figure 1S).”

In addition, “newly constructed vector” was changed to “AcrIIA4-Cdt1-2A-Cas9 expressing vector”.

The text associated with Fig S5 concludes that the decrease of NHEJ suggested that AcrIIA4-Cdt1 could be controlled by cell cycle. This is unclear. Are the authors comparing the 0.9% to 3.7% and saying that the fusion is limiting the activity of AcrIIA4 or comparing 8% to 3.7% saying that the decreased NHEJ shows AcrIIA4 is still working? This is unclear and somewhat misleading. If the latter, this conclusion is not warranted as the decreased NHEJ could just be due to AcrIIA4 activity being weakened by a fusion.

We agree that sentences caused misleading. We intended to mention the former case. We changed the sentences as follows (page 7, lines 15-22).

“Decrease of mutagenesis rates by NHEJ were exhibited when using vectors those have AcrIIA4-2A-Cas9 (0.9%) and AcrIIA4-Cdt1-2A-Cas9 (3.7%) compared to that of when using SpyCas9 alone (8.0%). The difference in decreased NHEJ rates between AcrIIA4 and AcrIIA4-Cdt1 suggests that the fusion is limiting the activity of AcrIIA4 by the cell-cycle dependent degradation of Cdt1. Expression of AcrIIA4-Cdt1 fusion showed higher mutagenesis rate compared to that of AcrIIA4 alone. It suggests that cell-cycle dependent expression of AcrIIA4-Cdt1 could inhibit SpyCas9 in the G1 phase and release it in the other phases while AcrIIA4 continuously inhibits SpyCas9.”

The experiment where precise editing of 1.6% increases to 2.0% should refer to Figure 2? Has significance testing been applied here? While the authors describe this as a slight increase (accurate), they conclude that this is “because of cell-cycle dependent activation of Cas9”. This claim is unwarranted as it is not associated with any proof that this is not assay variability or other factors. However, the maintained on-target with no off-target is very exciting. However, the authors should declare the limit of detection for the T7E1 assay.

Thank you for your helpful suggestion. We added a sentence to declare the limit of detection for the T7E1 assay (page 8, lines 9-10).

The changes in on-target HDR integration compared to the off-target are impressive. Are the authors comparing the % of on-target loci that have the HDR event to those that don't? are they still getting NHEJ in-dels? This should be clarified. And for off-target, this does not involve HDR, but is just an in-del mutation at a known Off target site, correct?

Given the impressive results I would suggest that the authors show representative gels from Fig 3 and 4 in the supplement.

HDR event was evaluated by digestion at restriction enzyme site inserted by HDR. It means we compared the % of on-target loci that have the HDR event to those that don't. T7E1 digestion detects total HDR and NHEJ events, which we referred as target NHEJ in the submitted manuscript. But to make it clear, "target NHEJ" was changed to "target T7E1" in Figure 6. For off-target, donor sequences don't have homology, so only in-del mutation should be detected.

The representative electrophoresis images were added as Figure S5.

Pertaining to fig 4, I don't actually see where in the text the authors discuss/refer to the actual AcrIIA4-Ctd1 data in Fig 4B/4C

Numbering of Figures 4B and 4C was changed to Figures 7B and 7C. We refer them in page 12, lines 8 and 10.

Discussion:

Second sentence of the discussion again mentioned the NHEJ decrease when AcrIIA4-Ctd1 was used and while this may be true, isn't it the case that this result alone doesn't prove it is due to Ctd1, since the AcrIIA4 protein alone also shows lower "NHEJ rate" (due to Cas9 inhibition).

We agree that the sentence was not clear. The sentence was changed to "The target mutation rate by NHEJ was decreased when SpyCas9-sgRNA and AcrIIA4-Cdt1 were co-expressed due to SpyCas9 inhibition by AcrIIA4" (page 12, lines 24-25).

Third sentence mentions dose-dependent inhibition. Where was this shown? Would be helpful to refer to the specific figures, from the discussion.

The sentence mentions about Figure 3. Figure information was added in p.12 line 26.

It would be helpful if the authors compare the HR rates they are getting (via plasmid or ssODN), during uninhibited Cas9 editing are comparable to other studies, especially if same sites have been used.

Thank you for your helpful suggestion. Unfortunately, as HDR rate using plasmid is very low or undetectable, comparison of HDR rate between the different donor types was not calculated. But it will be helpful analysis in future study. In addition, absolute values of editing efficiency may varies depending on the experimental conditions if comparing with the study from other groups. Thus, the

fold-increase of HDR rates would be the point to prove the effectiveness of our method.

Minor:

Adding line numbers would help the review process.

Sorry for your inconvenience. Page and line numbers were added to the main text.

Abstract; “these exhibited lower editing efficiency”... is this always true? insert “often”? or “sometimes”? Suggest “here, the anti-crispr protein...” Can make it more clear where this work starts. Typo “showed not only increased the frequency..” If ‘precise editing’ means HDR events, this should be articulated, to distinguish from off-target edits or off target HDR.

Thank you for your advices. Following your instruction, often was inserted to the sentence (page 1, line 21). In addition, “precise editing“ was changed to “homology-directed repair (or HDR)” through the main text.

Nice intro. Touches on the important points I expected to see

KD and KD both used. Please standardize. And I suggest you use ApoSpCas9 instead of just SpCas9 to distinguish from sgRNA-bound form.

Again, towards end of the intro, the authors should consider inserting a “here, we show” to distinguish between what is previous work, what would be hypothetically useful and then what they did.

Thank you for your suggestion. KD was integrated to K_D (page 2, line 30). SpyCas9 was changed to ApoSpyCas9 (page 2, line 30). The sentence “For activation in the S/G₂ phases and inactivation in the G₁ phase, anti-CRISPR was fused with N-terminal region of human Cdt1 (hCdt1)” was changed to “Here, we fused anti-CRISPR with N-terminal region of human Cdt1 (hCdt1) for activation in the S/G₂ phases and inactivation in the G₁ phase” (page 2, lines 35-36).

Results

AcrIIA4-Cdt1(30/120) nomenclature seems odd to me AcrIIA4-Cdt130-120?

AcrIIA4-Cdt1(30/120) was changed to AcrIIA4-Cdt1(30-120).

Thank you for your time and consideration. I look forward to hearing from you.

Sincerely,

Wataru Nomura, Ph.D.

REVIEWERS' COMMENTS:

Reviewer #1 (Remarks to the Author):

The authors have addressed my questions/concerns except for the major point I had raised about using the sequencing-based method for detection of indels and HDR. I have no further comments or requests besides grammatical and syntax edits.

Reviewer #2 (Remarks to the Author):

Abstract:

But also *suppress off-target effects

Intro:

Inserted text 8-10 seems to focus towards off-target effects, but real strength here is HDR outcomes, which are an independent and important problem. Authors could consider re-focusing this section

According to useful and important comments of the reviewers, we have appropriately prepared the second **REVISED** manuscript (COMMSBIO-20-0744-T). Correspondences to comments of reviewers 1 and 2 are described below point-by-point.

Comments of Reviewer #1

The authors have addressed my questions/concerns except for the major point I had raised about using the sequencing-based method for detection of indels and HDR. I have no further comments or requests besides grammatical and syntax edits.

Thank you very much again for your review of our revised manuscript.

About sequencing for mutagenesis rate calculation, again, as far as we analyze the efficiency with the same method, we think relative changes of efficiency are valid. However, we are aware of importance of evaluation methods other than T7E1 assay, we have added sentences that mention the detection limit of T7E1 assay and effective methods for evaluation of editing efficiency with citation of the recommended articles (page 8, lines 9-13).

English language editing was done again on the revised manuscript by Editage (www.editage.com).

Comments of Reviewer #2

Abstract:

But also *suppress off-target effects

Thank you very much again for your review of our revised manuscript.

The part of abstract was changed as suggested.

Intro:

Inserted text 8-10 seems to focus towards off-target effects, but real strength here is HDR outcomes, which are an independent and important problem. Authors could consider re-focusing this section.

Thank you for a useful suggestion. In this paragraph, we mainly focused on the off-target effects. HDR outcomes are discussed in the next paragraph. However, as reviewer suggested, HDR outcomes are independent and important problem. Thus, in this paragraph, we further added a phrase “in addition to endeavour to increase the efficiency of precise editing at on-targets” in line 11, page 2. This addition would emphasize the importance of HDR outcomes.

Thank you for your time and consideration. The manuscript was improved by reviewers' critical and constructive suggestions. I look forward to hearing from you.

Sincerely,

Wataru Nomura, Ph.D.